# Structural Equation Modeling of Tongue Function and Tongue Hygiene in Acute Stroke Patients

**DOI:** 10.3390/ijerph18094567

**Published:** 2021-04-26

**Authors:** Rena Hidaka, Junichi Furuya, Akira Nishiyama, Hiroyuki Suzuki, Michiyo Aoyagi, Chiaki Matsubara, Yu Yoshizumi, Kanako Yoshimi, Ayako Nakane, Haruka Tohara, Yuji Sato, Shunsuke Minakuchi

**Affiliations:** 1Oral Health Sciences for Community Welfare, Graduate School of Medical and Dental Sciences, Tokyo Medical and Dental University (TMDU), Tokyo 113-8510, Japan; n-rena.ohcw@tmd.ac.jp; 2Geriatric Dentistry, Showa University School of Dentistry, Tokyo 145-8510, Japan; sato-@dent.showa-u.ac.jp; 3Dental Anesthesiology and Orofacial Pain Management, Graduate School of Medical and Dental Sciences, Tokyo Medical and Dental University (TMDU), Tokyo 113-8510, Japan; anishi.tmj@tmd.ac.jp; 4Gerodontology and Oral Rehabilitation, Graduate School of Medical and Dental Sciences, Tokyo Medical and Dental University (TMDU), Tokyo 113-8510, Japan; h.suzuki.gerd@tmd.ac.jp (H.S.); m.chiakingyo@gmail.com (C.M.); s.minakuchi.gerd@tmd.ac.jp (S.M.); 5Dysphagia Rehabilitation, Graduate School of Medical and Dental Sciences, Tokyo Medical and Dental University (TMDU), Tokyo 113-8510, Japan; sacra333march@gmail.com (M.A.); k.yoshimi.gerd@tmd.ac.jp (K.Y.); a.nakane.swal@tmd.ac.jp (A.N.); h.tohara.swal@tmd.ac.jp (H.T.); 6Oral Surgery, Saitama Red Cross Hospital, Japanese Red Cross Society, Saitama 330-0081, Japan; y.y.zumi@gmail.com

**Keywords:** structural equation modeling, tongue function, tongue hygiene, oral health assessment tool, acute stroke

## Abstract

In acute stroke patients, it is important to maintain tongue hygiene and tongue function for prognosis management. However, the direct relationship between these factors is unclear, since these are often assessed by multiple observables. In this study, we used structural equation modeling (SEM) analysis, a tool to analyze the relationship between concepts that cannot be measured directly, to analyze the relationship between tongue hygiene and tongue function. The subjects were 73 patients with acute stroke admitted to a university hospital who underwent dental intervention. Age, sex, nutritional intake method, clinical severity classification of dysphagia, number of current teeth, number of functional teeth, oral health, tongue movement, tongue coating, number of microorganisms on the tongue surface, tongue surface moisture level, and tongue pressure were measured at the first visit. SEM analysis showed that the relationship between tongue function and tongue hygiene was 0.05 between tongue function and swallowing function was 0.90, and that between tongue hygiene and swallowing function was 0.09. We found no statistical relationship between tongue function and tongue hygiene in acute stroke patients. However, it was reconfirmed that tongue function is strongly related to feeding and swallowing functions.

## 1. Introduction

Patients with acute stroke have a high incidence of dysphagia. Swallowing function is controlled by both brain hemispheres. However, the dominant hemisphere for controlling swallowing function is not known. If the right hemisphere is affected, it may cause aspiration and pharyngeal phase dysfunction, and if the left hemisphere is affected, oral phase dysphagia may result [1]. According to previous studies, dysphagia is associated with a higher risk of aspiration pneumonia [2,3] and the development of aspiration pneumonia is associated with longer hospital stays and higher mortality [4,5]. Marik et al. [6] listed “risk factors for aspiration” as “neurogenic dysphagia,” “disorders of the esophagogastric junction,” and “anatomical abnormalities of the upper airway or upper gastrointestinal tract.” Aspiration of saliva containing oral microorganisms is an important factor in the development of aspiration pneumonia [7]. Aspiration of oral bacteria due to decreased swallowing and coughing reflexes associated with cerebrovascular disease can cause aspiration pneumonia [8]. In addition to dysphagia, several other factors are thought to be involved, including the number of cavities, smoking, oral intake, and oral care [9,10]. The risk of developing aspiration pneumonia is reported to be increased not only by dysphagia but also by poor oral hygiene. Previous studies have suggested a direct relationship between the results of visual assessment of tongue lichen in edentulous elderly people and the number of microorganisms in saliva and the development of aspiration pneumonia, suggesting that poor oral hygiene, especially tongue hygiene, increases the number of bacteria, leading to poor oral health [8,11]. Obana et al. reported that acute stroke patients have poor tongue, denture, and oral cleanliness [12].

It has also been reported that stroke causes tongue function to decline, making oral intake difficult and leading to poor oral hygiene and aspiration pneumonia [13]. A previous study used tongue pressure measurement as a predictor for the development of aspiration pneumonia in acute stroke patients. When the measurements taken in patients with low tongue pressure at the time of initial assessment were compared with measurements after two weeks, it was found that the frequency of aspiration onset was higher in the non-improved group than in the group with improved tongue pressure [14]. A study of elderly people suggested that there was a correlation between the degree of tongue lichen adherence and decreased tongue motor function [15]. These reports suggest that maintaining and improving the function of the tongue may reduce the risk of aspiration pneumonia.

We also explored factors related to the number of bacteria on the back of the tongue and the degree of tongue lichen adherence as indicators of tongue hygiene and found that the degree of tongue lichen adherence and the degree of wetness on the back of the tongue were related to the increase in the number of bacteria on the back of the tongue [16]. However, the observed variables representing oral hygiene and tongue motor function differed among studies, and it is unclear whether there is a direct relationship between the factors indicated by these results. In this study, we used dysphagia, which is a risk factor for aspiration pneumonia, as an index to examine whether tongue function and tongue hygiene are more related to the development of dysphagia than other factors in acute stroke patients by conceptualizing each measurement item.

Structural equation modeling (SEM) analyzes the structure of covariance between such observed variables and compares concepts that cannot be directly observed, such as tongue movement function, to examine the structure of causal relationships. Until now, SEM analysis for the elderly has been dominated by reports on oral-related quality of life and the difference between subjective and objective evaluations related to oral hygiene [17,18,19,20], and no reports on oral hygiene status and oral function are available to the best of our knowledge. Therefore, the purpose of this study was to clarify the relationship between tongue hygiene and tongue function using SEM analysis.

## 2. Materials and Methods

### 2.1. Participants

A total of 207 acute stroke patients with subarachnoid hemorrhage, cerebral infarction, or cerebral hemorrhage not caused by trauma, who were referred to the Outpatient Department of Dysphagia Rehabilitation, Dental Hospital, Tokyo Medical and Dental University from 1 April 2016, to 31 March 2019, and whose attending physicians requested dental intervention, were included in the study. We excluded 134 subjects who had not given consent, or those in whom measurement was difficult to perform, such as those who were intubated or clenched and could not open their mouths and be swabbed to obtain bacteria.

This study was conducted with the approval of the Ethical Review Committee of the School of Dentistry, Tokyo Medical and Dental University (Approval No. D2015–503). The study conformed to the tenets of the Declaration of Helsinki (2013). All study participants were fully informed, verbally and in writing, of the study protocol, and they provided written consent.

### 2.2. Study Protocol

At the first visit for dental intervention, one dentist and one dental hygienist performed a systemic and intraoral evaluation of the patient. The systemic evaluation included age, sex, nutritional intake method (Functional Oral Intake Scale (FOIS)), and clinical severity of dysphagia (Dysphagia Severity Scale: DSS). The FOIS includes eight levels of nutritional intake method, from level 1 (tube feeding only, no oral intake) to level 7 (oral nutrition with no specific restrictions (regular diet)) [21]. DSS is an eight-point scale ranging from 1 (salivary aspiration: aspiration of everything, including saliva, with poor respiratory status or no gag reflex) to 7 (normal range: no clinical problems) [22]. In the oral assessment, the following eight items were evaluated: number of current teeth, number of functional teeth, comprehensive assessment (Oral Health Assessment Tool: OHAT) [23], tongue movement, tongue coating index (TCI), number of microorganisms on the tongue surface, tongue surface moisture, and tongue pressure. The OHAT is a comprehensive intra-oral environment assessment tool that can be easily used by non-dentistry professionals. Based on previous studies [23], eight items (lips, tongue, gingiva/mucosa, saliva, remaining teeth, dentures, oral cleaning, and toothache) were evaluated on a 3-point scale, namely healthy (0 points), slightly poor (1 point), or diseased (2 points), according to the criteria. The total of the scores for each item was used as the total OHAT score. Tongue movement disorder was assessed by evaluating the anterior-posterior movement of the tongue on a 4-point scale (0 = beyond the lower lip, 1 = up to the lower lip, 2 = does not reach the lower lip, 3 = no movement possible) [24]. The TCI of Winkel et al. [25] was used to determine the degree of tongue lichen adhesion. This evaluation method divides the tongue into 6 regions (3 anterior and 3 posterior) and uses the total score of the degree of tongue lichen adhesion (0 = no adhesion, 1 = thin lichen adhesion that does not cover the tongue papillae, 2 = thick adhesion that covers the tongue papillae) [25]. A bacterial counter (PHC, Japan) was used to measure the number of bacteria on the tongue surface. The microorganism numbers on the tongue surface were classified into seven levels––LV1: <105 CFU/mL, LV2: 105–106 CFU/mL, LV3: 106–106.5 CFU/mL, LV4: 106.5–107 CFU/mL, LV5: 107–107.5 CFU/mL, LV6: 107.5–108 CFU/mL, and LV7: >108 CFU/mL. For sample collection, a sterile cotton swab was moistened with water and brushed thrice in each direction over a 1-cm distance on the central part of the tongue surface with a pressure of 20 g [26]. The swab was placed in the bacterial counter where the number of microorganisms was determined. An oral moisture meter, Mucus (Life, Japan), was used to measure the degree of wetness. Measurements were taken by pressing vertically on the back of the tongue, 10 mm from the tip of the tongue with a pressure of approximately 200 g. Measurements were taken three times, and the median value was used. Tongue pressure was measured using a tongue pressure-measuring device (JMS, Japan). The patient was instructed to press the balloon of the tongue-pressure probe against the hard palate with the tip of the tongue using maximum pressure, and measurements were taken after practice. Measurements were taken three times, and the average value was used as the maximum tongue pressure.

### 2.3. Statistical Analysis

Spearman’s correlation coefficients among the survey items were calculated. From the factors with significant correlation, items that were considered highly related to tongue hygiene and tongue function were selected and subjected to factor analysis (principal factor method, Promax rotation). Based on previous studies [10,11,12,13,14,15,16], we hypothesized that decreased tongue function would be related to dysphagia and poor tongue hygiene. We then conducted a path analysis of the relationship between each concept to test our hypothesis. The goodness of fit index (GFI), adjusted goodness of fit index (AGFI), and root mean square error of approximation (RMSEA) were used as fit indices for the tongue function model. The model was considered fit when the GFI and AGFI were greater than 0.9 and RMSEA was less than 0.05. SPSS version 25 (IBM Corp., Armonk, NY, USA) was used for all statistical analyses. The significance level was set at 0.05.

## 3. Results

Out of the 207 subjects, the final study population consisted of 73 subjects (49 men and 24 women, with a mean age of 65.4 ± 14.8 years). The mean, standard deviation, and median values of each measurement are shown in Table 1. Most of the subjects were not receiving oral intake and had some oral problems. Tongue movement was not sufficient, and tongue pressure was low. Most of the patients had lichen on their tongue. The items that showed correlation with tongue movement were OHAT (*r* = 0.274, *p* = 0.019), tongue surface moisture (*r* = −0.549, *p* = 0.000), tongue pressure (*r* = −0.674, *p* = 0.000), DSS (*r* = −0.692, *p* = 0.000), and FOIS (*r* = −0.474, *p* = 0.000) (Table 2). The items that correlated with TCI were oral moistness (*r* = −0.269, *p* = 0.021) and tongue pressure (*r* = −0.247, *p* = 0.036). There were no items that correlated with the number of bacteria on the back of the tongue.

DSS, FOIS, tongue movement, tongue pressure, TCI, and bacterial count were identified as factors related to tongue function and tongue hygiene, and factor analysis was conducted. The results divided the factors into two groups. Group 1 included DSS, FOIS, tongue movement, and tongue pressure, and Group 2 included TCI and bacterial count. After examining the content of the groups, we finally classified them into three component concepts: tongue function, swallowing function, and tongue hygiene (Table 3, Figure 1). The results of the path analysis of the relationships among these concepts are shown in Figure 2. The GFI and AGFI were both above 0.9, and the RMSEA was below 0.05, indicating a good fit of the model. The relationship between tongue function and tongue hygiene was 0.05, between tongue function and swallowing function was 0.90, and that between tongue hygiene and swallowing function was 0.09.

## 4. Discussion

The purpose of this study was to clarify the relationship between tongue function and tongue hygiene in acute stroke patients by SEM analysis. In the present study, tongue movement and tongue pressure were used as indicators of tongue function. The number of bacteria on the back of the tongue and TCI were used as indicators of tongue hygiene. We found that there was no direct relationship between tongue function and tongue hygiene.

In this survey, we included only items that were considered particularly related directly to tongue function and tongue hygiene. According to a previous study, the amount of tongue moss adhesion was increased when tongue pressure decreased, whereas a significant decrease in tongue moss was observed after oral function training [15]. According to Ralph et al. [27], elderly people are more prone to tongue lichen adherence than younger people are due to changes in eating habits, poor oral hygiene, decreased saliva volume, and altered saliva properties. In the present study, we used tongue movements and tongue pressure as indicators of tongue function. Tongue pressure has been reported to decrease in people over 70 years of age, and the scores for tongue lichen adherence are higher in people over 80 years of age than in those below 80, which suggest that both are affected by aging [28]. The mean age of the subjects in this study was 65.4 years (SD 14.8). There were no items that correlated with age in our study, and we can assume that the effect of aging on tongue function and tongue hygiene, the purpose of this study, may not be significant. However, since it has been shown that tongue function declines with age, it is necessary to intervene at an early stage to prevent the decline in tongue function when it is likely to decline due to disease symptoms, such as in acute stroke patients.

It has been reported that the number of oral bacteria increases in cancer and cardiac patients when they do not consume orally [29]. Oral intake is thought to indicate that a certain level of oral function is maintained. Murray et al. [30] reported that in post-stroke patients with dysphagia (mean 19.8 days post-onset) and without dysphagia (mean 41.1 days post-onset), OHAT scores were 4 and 2, respectively, at the time of the first intervention, whereas the OHAT scores were 3 and 2, respectively, at the time of the second intervention. This suggests that there may be a tendency for the OHAT score to improve or not deteriorate, after 1 week of oral hygiene management_._ The present study was conducted within 2 weeks after the onset of illness, which is a shorter period compared to their study and resulted in slightly higher OHAT values. The scores for oral dryness and oral cleaning were generally high, suggesting that oral hygiene deteriorated because of the effects immediately after onset. However, due to the shorter period after the onset of illness in the current study, the impact on the decline in oral function due to the onset of poor oral hygiene status may have been small, and thus, no association was found.

In this study, we did not include the level of consciousness in the path analysis. The DSS and FOIS were used to examine feeding and swallowing function; the DSS was judged by the dentist after considering the level of consciousness and other factors. Therefore, we did not include these as separate items because we believed that the results reflected the patient’s level of consciousness and other factors.

There are reports that the risk of respiratory-infection-causing bacteria becoming established during the recovery period is increased [31]. The environment on the dorsum of the tongue is particularly conducive to bacterial growth when self-cleaning is impaired by inhibited swallowing. In the present study, most of the patients had lichen on their tongue [32,33,34,35]. It has been reported that the adherence of tongue moss is correlated with the biofilm on the tongue and halitosis; good tongue hygiene may reduce the number of bacteria on the back of the tongue and help prevent aspiration pneumonia. It has also been reported that oral health of patients in the acute phase of stroke can be significantly improved through collaboration between medical and dental care i.e., interdisciplinary oral health care, in which nurses, speech therapists, dentists, and dental hygienists work together [12]. In this study, we found no direct relationship between tongue function and tongue hygiene and a strong relationship between feeding and swallowing functions and tongue function. Previous studies have shown that age-related dysphagia is associated with a decrease in tongue and jaw opening strength in healthy elderly men and a decrease in tongue strength in elderly women, suggesting that a decrease in tongue muscle strength and jaw opening strength may affect feeding and swallowing functions [35,36]. It has been shown that in acute stroke patients, individualized rehabilitation from an early stage can effectively improve the symptoms of dysphagia [37]. In order to restore the swallowing function and implement oral intake from an early stage, it is suggested that interventions focusing on the maintenance and improvement of tongue function, as well as improvement of tongue hygiene, may be necessary in acute stroke patients.

One of the limitations of this study is that, although we included all hospitalized patients, we excluded those who were intubated or who were unable to open their mouths because of clenching and could not be swabbed for bacteria. In addition, we did not classify the patients according to stroke subtype or severity, and we did not investigate the characteristics of each type. In addition, we did not investigate the incidence of aspiration pneumonia, however, we plan to collect more information in further studies.

## 5. Conclusions

We investigated the relationship between tongue hygiene and tongue function in acute stroke patients and found no statistical relationship. However, it was reconfirmed that tongue function is strongly related to feeding and swallowing functions.

## Figures and Tables

**Figure 1 ijerph-18-04567-f001:**
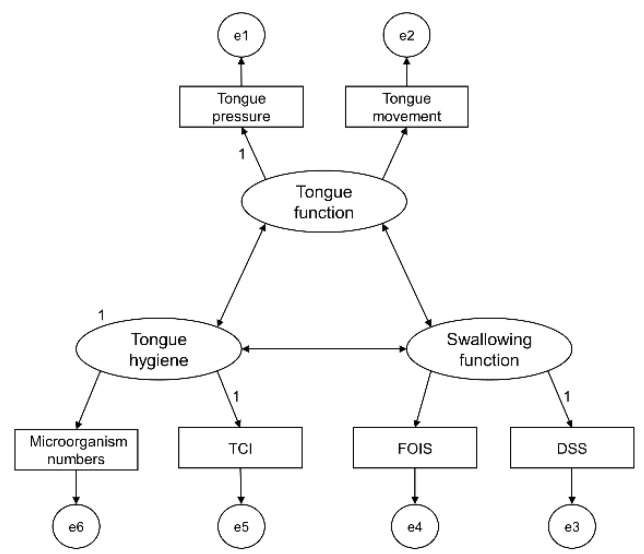
Hypothesized structural model. DSS, Dysphagia Severity Scale; FOIS, Functional Oral Intake Scale; TCI, tongue coating index. e1 to e6 are error variables.

**Figure 2 ijerph-18-04567-f002:**
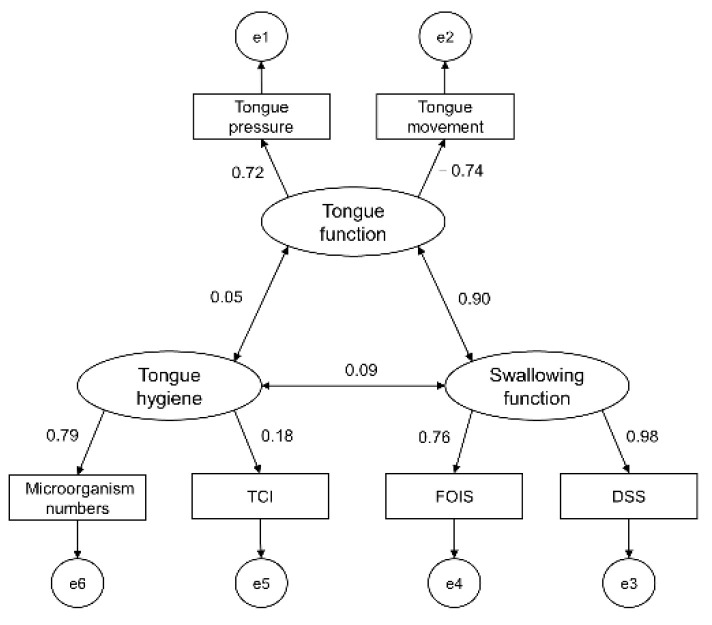
Standardized path coefficients GFI = 0.977, AGFI = 0.932, RMSEA = 0.000; el to e6 are error variables. Tongue function: tongue hygiene: swallowing function. GFI, goodness of fit index; AGFI, adjusted goodness of fit index; RMSEA, root mean square error of approximation.

**Table 1 ijerph-18-04567-t001:** Patient characteristics and results of each outcome.

Measurement Items	Mean (SD)	Median (25, 75)
DSS	2.75 (1.76)	2.00 (1.0, 4.0)
FOIS	2.14 (1.96)	1.0 (1.0, 2.5)
Number of functional teeth	23.22 (7.68)	27.0 (20.5, 28.0)
OHAT total score	4.85 (2.62)	5.0 (3.0, 6.5)
Tongue surface moisture level	18.72 (11.39)	23.5 (7.2, 28.1)
Tongue movement	1.44 (1.44)	1.0 (0.0, 3.0)
Tongue pressure (kPa)	8.73 (11.57)	2.3 (0.0, 13.3)
TCI	6.79 (3.29)	6.0 (4.0, 9.0)
Microorganism numbers on the tongue surface (LV)	4.29 (1.27)	5.0 (3.0, 5.0)

DSS, Dysphagia Severity Scale; FOIS, Functional Oral Intake Scale; TCI, tongue coating index; OHAT, Oral Health Assessment Tool; LV, level.

**Table 2 ijerph-18-04567-t002:** Correlation between the factors.

Measurement Items	1	2	3	4	5	6	7	8	9	10	11
1. Age	1										
2. Sex	0.084	1									
3. Tongue movement	0.059	0.149	1								
4. Tongue pressure	−0.145	−0.148	−0.674 **	1							
5. TCI	−0.155	−0.079	0.105	−0.247 *	1						
6. Microorganism numbers	−0.038	−0.172	−0.070	−0.023	0.183	1					
7. DSS	−0.011	−0.259	−0.692 **	0.605 **	0.057	0.037	1				
8. FOIS	0.084	−0.122	−0.474 **	0.479 **	0.133	0.026	0.694	1			
9. OHAT	0.155	−0.124	0.274 *	−0.261 *	0.208	0.057	−0.259 *	−0.122	1		
10. Number of functional teeth	−0.207	−0.226	0.065	0.159	−0.107	−0.123	0.196	0.195	−0.511 **	1	
11. Moisture level	−0.058	−0.094	−0.549 **	0.442 **	−0.269 *	0.054	0.528 **	0.489 *	−0.301 **	0.240	1

* *p* < 0.5, ** *p* < 0.01. DSS, Dysphagia Severity Scale; FOIS, Functional Oral Intake Scale; TCI, tongue coating index; OHAT, Oral Health Assessment Tool.

**Table 3 ijerph-18-04567-t003:** Factor analysis results.

Concepts	Items	Factor 1	Factor 2
Swallowing function	DSS	0.958	0.045
	FOIS	0.760	0.080
Tongue function	Tongue movement	−0.698	−0.032
	Tongue pressure (kPa)	0.690	−0.148
Tongue hygiene	TCI	−0.120	0.481
	Microorganism numbers on the tongue surface (LV)	0.138	0.353

DSS, Dysphagia Severity Scale; FOIS, Functional Oral Intake Scale; TCI, tongue coating index.

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
