# Peer review of "Structural Equation Modeling of Tongue Function and Tongue Hygiene in Acute Stroke Patients"

_ijerph, 2021, doi:10.3390/ijerph18094567_

Round 1
Reviewer 1 Report
Thank you very much for submitting this very interesting article.
I would appreciate if the authors can address the following questions.
- It states that 134 subjects were excluded from the study. That represent a significant number of the 207 asked to participate. It states that patients were excluded due to consent but also in "those whom measurement was difficult to perfrom.?. CAn the author elaborate more on what factors make measurement more difficult to perform and measure?. In addition do the authors feel that excluding 1/3 of subjects may have skewed the data? Due you think there was a specific type of stroke patient that may have been eliminated from the analysis
2. Can the authors identify how long after the incidental stroke that the evaluations were done? What is the acute period and was this a one time evaluation?? Did you look at function of patients with stroke just affecting the limbs and those with involved facial muscles.?
3. Do the authors have an analysis of stroke subtype and etiology or NIHSS or stroke severity in order to ascertain if this is a variable should be further explored.
4. Do the authors have data on which of the patients developed aspiration pneumonia in their cohort. Do you have antibiotic use or associated initial infection that may have been treated that may alter "Bacteria counts"
5. Does the bacterial counter give information as to the type of bacteria. Did you evaluate bacteria type present? If so were any of them associated with possible stroke (i.e chlamydia)
6.
Reviewer 2 Report
The abstract does not correspond to the development of the study
The introduction is incomplete, it needs a good bibliographic review that allows to justify with evidence why and why of this study.
There is no PICO question or structure of it. The objective of the study and the research question are not indicated
It is indicated that a variable relationship is going to be made with aspiration pneumonia, and this relationship is not made
Results tables are not explained
The inclusion and exclusion criteria of the study are not indicated.
Who are evaluators is not indicated
The characteristics of the evaluation tools are not indicated
The discussion is not supported by evidence
There are no conclusions
Reviewer 3 Report
The authors present a paper regarding the structural equation modeling
of tongue function and tongue hygiene in acute stroke patients. the
methods are appropriate and well described, and adequate details are
provided to replicate the work. The discussion and conclusion are well
balanced and adequately supported by the data.
The paper is clearly written and I believe it should be accepted after some
minor revisions: the materials and methods should be improved: 1)
adding information about the bacterial counter used (line
122; study protocols); 2) please note the causes of dysphagia are not the
decreased tongue functioning, but the nervous low-functionality of the ninth and tenth cranial nerves (line 138 - Statistical analysis); 3) in the
lines 111 and 137 the sentence "Based on previous studies...." are not
supported by bibliography: add study references.
Discussion: it would add value also the role of tongue biofilm as
causative agent of other conditions such as Halitosis. I suggest these
articles as useful to find such info:
https://pubmed.ncbi.nlm.nih.gov/27789913/ ;
https://pubmed.ncbi.nlm.nih.gov/31725203/ ;
https://pubmed.ncbi.nlm.nih.gov/31034083/ )
Round 2
Reviewer 2 Report
Thank you for making the indicated changes. Thus the study is more enriching. In the introduction, it would be necessary to specify causes of dysphagia, causes of aspiration pneumonia, and from there include the function of the tongue.
The bibliographic review needs to be more exhaustive, to improve the introduction and the discussion.
Author Response
Thank you for your suggestion. We have revised the Introduction and Discussion accordingly.